# The Crucial Roles of Pitx3 in Midbrain Dopaminergic Neuron Development and Parkinson’s Disease-Associated Neurodegeneration

**DOI:** 10.3390/ijms24108614

**Published:** 2023-05-11

**Authors:** Xin Wang, Xi Chen, Guangdong Liu, Huaibin Cai, Weidong Le

**Affiliations:** 1Institute of Neurology, Sichuan Provincial People’s Hospital, University of Electronic Science and Technology of China, Chengdu 611731, China; wangxinwx109@163.com (X.W.);; 2Chinese Academy of Sciences Sichuan Translational Medicine Research Hospital, Chengdu 611731, China; 3Transgenic Section, Laboratory of Neurogenetics, National Institute on Aging, National Institutes of Health, Bethesda, MD 20892, USA

**Keywords:** Parkinson’s disease, paired-like homeodomain transcription factor 3 (Pitx3), midbrain dopaminergic neuron development, transcription factors

## Abstract

The degeneration of midbrain dopaminergic (mDA) neurons, particularly in the substantia nigra pars compacta (SNc), is one of the most prominent pathological hallmarks of Parkinson’s disease (PD). To uncover the pathogenic mechanisms of mDA neuronal death during PD may provide therapeutic targets to prevent mDA neuronal loss and slow down the disease’s progression. Paired-like homeodomain transcription factor 3 (Pitx3) is selectively expressed in the mDA neurons as early as embryonic day 11.5 and plays a critical role in mDA neuron terminal differentiation and subset specification. Moreover, Pitx3-deficient mice exhibit some canonical PD-related features, including the profound loss of SNc mDA neurons, a dramatic decrease in striatal dopamine (DA) levels, and motor abnormalities. However, the precise role of Pitx3 in progressive PD and how this gene contributes to mDA neuronal specification during early stages remains unclear. In this review, we updated the latest findings on Pitx3 by summarizing the crosstalk between Pitx3 and its associated transcription factors in mDA neuron development. We further explored the potential benefits of Pitx3 as a therapeutic target for PD in the future. To better understand the transcriptional network of Pitx3 in mDA neuron development may provide insights into Pitx3-related clinical drug-targeting research and therapeutic approaches.

## 1. Background

Parkinson’s disease (PD) is the second most frequent age-dependent neurodegenerative disorder after Alzheimer’s disease and is estimated to affect more than 6 million people worldwide [1]. The typical clinical manifestations of PD include resting tremors, abnormal gait (shuffling and reduced step length), bradykinesia and rigidity, as well as nonmotor symptoms (hyposmia, constipation, and sleep behavioral disorder) [2,3]. These motor/non-motor symptoms could trigger widespread disability among PD patients [3]; therefore, how to slow down the disease progression to improve the quality of life in PD patients deserves more attention.

The main pathological feature of PD is the progressive loss of midbrain dopaminergic (mDA) neurons, which is possibly caused by mitochondrial dysfunction, the disruption of vesicle trafficking, autophagy impairment, or/and the defective proteasomal clearance of misfolded proteins [4,5]. These pathogenic effects, individually or collectively, lead to the apoptotic (programmed) cell death of mDA neurons in the end [6].

mDA neurons in the SNc and VTA of the midbrain are the primary DA source in the mammalian central nervous system [7]. During development, the precursors of DA neurons migrate from the neuroepithelium to the ventral midbrain. Sonic hedgehog (Shh) and fibroblast growth factor (Fgf8), two diffusible factors, collaborate to promote neuronal development. Lmx1a and Msx1 appear early, while precursor cells proliferate, and these are essential for the passage of the proliferating precursor cells into the first stages of differentiation into mDA neurons [8]. Lmx1b, the engrailed factors En1 and En2, Foxal, and other factors are also involved in the early stage of mDA-specific differentiation (at E9-E10 in mice) when the first gene for DA synthesis, amino acid decarboxylase (AADC), is induced. Between E9.5 and E13.5 in mice, midbrain DA progenitors exit the cell cycle and enter post-mitosis [9]. Then, at E10.5 and E11.5, the transcription factors for terminal differentiation are activated by Nurr1 and Pitx3, respectively. Nurr1 is required for the induction of tyrosine hydroxylase (TH) at E11.5, vesicular monoamine transporter 2 (Vmat2) at E12.5, and dopamine transporter (DAT) at E14 [10], and Pitx3 is necessary for some terminally differentiated SNc DA neurons to survive [7].

mDA neurons are generally divided into three main clusters, which are located in substantia nigra pars compacta (SNc), ventral tegmental area (VTA), and the retrorubral field (RRF). SNc mDA neurons primarily project their axons to the dorsolateral striatum, a component of the basal ganglia circuitry, and DA released from SNc mDA neurons mediates the activity of targeted striatal cells to regulate voluntary movements and motor learning [11]. In contrast, VTA mDA neurons mainly innervate the ventromedial striatum (nucleus accumbens, NAc), prefrontal cortex (PFC; mesocortical pathway), and limbic regions (mesolimbic network). DA release from VTA and RRF mDA neurons are involved in the modulation of cognitive, emotive/affective, motivational/salient, and rewarding behaviors [11]. All three types of mDA neurons undergo multiple developmental stages, including the patterning, induction, specification of mDA progenitors, and mDA neuron maturation, which are controlled by numerous genetic networks and cellular signaling pathways [12,13]. The distribution and projections of mDA neurons are detailed in Figure 1.

Currently, the main PD treatments still focus on increasing the DA level or regulating DA transmission via pharmacotherapy [14]. Moreover, patients who have noticeable tremors or uncontrollable motor fluctuations may also benefit from a deep brain stimulation operation [15]. However, instead of preventing or delaying the progression of the disease, these treatments can only partially mitigate the physical symptoms [16]. Meanwhile, substantial side effects, such as dyskinesia and impulse control difficulties, may appear after long-term pharmacological treatments [17]. Cell transplantation, a potential strategy used to replace impaired mDA neurons, has gained particular interest. It is known that some transcription factors are crucial for the growth, differentiation, specification, and transmitter synthesis of mDA neurons, including forkhead box A2 (Foxa2), LIM homeobox transcription factor1 A (Lmx1a), nuclear receptor-related factor1 (Nurr1), paired-like homeodomain3 (Pitx3), aldehyde Dehydrogenase 1 family member A1 (Ascl1), neurogenin2 (Ngn2), and neuronal differentiation 1 (NeuroD1) [16]. Pitx3 and Nurr1 are mdDA neuron-specific markers, whereas Ascl1, Ngn2, and NeuroD1 are markers for neural progenitor cells. Additionally, Neurotrophic factors, including glial-cell-derived neurotrophic factor (GDNF) and brain-derived neurotrophic factor (BDNF), also play a crucial part in the differentiation, development, and maintenance of mDA [16,18,19,20,21,22]. Thus, expanding the knowledge of these transcription factors and neurotrophic factors in neuronal formation will greatly benefit the generation of functional mDA neurons that are applied to cell replacement therapy for PD patients.

Among them, Pitx3 is mainly expressed in mDA neurons and plays an essential role in mDA neuronal development [23,24]. The expression of Pitx3 is known to initiate as early as embryonic day 11.5, and it is distributed in the lens, skeletal muscles, and mDA neurons during early development [23,25,26]. However, postnatally, its expression is preserved only in postmitotic mDA precursors and mature mDA neurons particularly in SN, suggesting a key role for Pitx3 in the maintenance of mDA neuronal function [23,24,27]. Pitx3-deficient Aphakia (*ak*) mice exhibit the preferential loss of SNc mdDA neurons during embryonic and postnatal development, reiterating that Pitx3 plays a critical role in early mdDA neuronal events. Additionally, the expression of Pitx3 was reported to be significantly downregulated in both PD patients’ peripheral blood lymphocytes and brain tissues [25,28]. Additionally, several Pitx3 gene variants have been identified to be linked to sporadic PD [29,30]. Thus, these data altogether establish the key role of Pitx3 not only in early mDA development, but also in retaining the normal physiological functions in adult neurons.

Pitx3 has the highest level of expression in mDA neurons among these transcription factors at the embryonic stages [31,32]. How this relatively abundant protein binds to other transcription factors to coordinate the mDA neuronal development is not clear yet. In this review, we focus on the discussion of Pitx3 and its crosstalk with other transcription factors during mDA neuronal specification and differentiation. Furthermore, the potential therapeutic role of Pitx3 is also summarized. Our study may help readers to further understand the Pitx3-centered neural transcription factor network and provide insights on how to improve future PD treatments.

## 2. Introduction of Pitx3

Pitx3, a member of the pituitary homeobox subfamily, is a transcription factor containing homeodomains located on chromosome 10q24 [33]. The first member of the pituitary homeobox family to be cloned was Pitxl (previously known as Ptx1). It was found to support pituitary development and function [34]. Shortly after a second family member, Pitx2 (called Ptx2 or Rieg1), was discovered to be involved with Rieger’s syndrome, causing a deformity of craniofacial characteristics [35] Then, Pitx3 (Plx3) was identified during embryogenesis. It is momentarily expressed in skeletal muscle and the eye lens [23], but after birth, the content of Pitx3 is highly restricted, and it is constitutively expressed in the SNc and VTA of the midbrain [23].

### 2.1. Pitx3 Structure

In humans, Pitx3 has four exons and is found on chromosome 10. The protein is 302 amino acids long, and the cDNA is 1407 bp in length [26]. The mouse Pitx3 gene also has four exons and is located on chromosome 19. cDNA has a length of 1379 bp, and the protein is also 302 amino acids long [25,36]. The pitx3 genes of humans and mice are highly homologous and nearly 90% identical at the nucleotide level throughout the coding region [36].

### 2.2. Pitx3 Protein

Pitx3 coordinates gene activity that determines cell destiny during development in a variety of species, including yeasts and mammals [25]. The protein has an OAR domain at its C-terminus, which is present in several types of paired-like homeodomains [31]. Although the function of OAR in the Pitx3 protein is uncertain, OAR’s impact on DNA binding appears to limit the transcriptional activity of Cartl protein. The OAR domain and N-terminus of the Pitx and Prx OAR-related proteins are thought to intramolecularly interact, resulting in a protein conformation that is connected to a generally inactive state of these transcription factors [37].

### 2.3. Pitx3 in the Main Subgroups of mDA Neurons

Pitx3 is a transcription factor required for the development and survival of mdDA neurons [38]. In the midbrain, DA neurons have two primary subgroups: the SNc and the VTA [7]. Although most mDA neurons express Pitx3, the Pitx3 deficiency in mice only impacts a portion of the mDA neurons. These findings show that Pitx3 may play distinct roles in these two midbrain subregions [25]. Pitx3 and TH are both expressed in the VTA at approximately the same time, and TH is still expressed in VTA even in the absence of Pitx3. However, in SNc, Pitx3 is expressed before TH is, and the SNc precursors are unable to generate TH without Pitx3 [39]. Thus, the VTA DA neurons in Pitx3-deficient newborn mice remain unaffected, whereas SNc neurons have been significantly lost potentially due to decreased TH levels [40,41]. These findings imply that Pitx3 is crucial for maintaining normal function and preserving the specific molecular identities of postnatal mDA neurons [42].

## 3. Pitx3 and Pitx3-Related Transcriptional Networks in mDA Neuron Development

The mDA neuronal population is unevenly distributed during early developmental stages, where it divides into several subpopulations with distinct topographical and temporal expression patterns [43]. Pitx3 is a key player in subset specification and is required for the proper survival of an mDA neuron subset in prenatal and postnatal mouse brains [44]. Based on its crucial role in mDA neurons, the interaction network of Pitx3 with other important transcription factors is also elucidated, which may contribute to a better understanding of mDA neuronal differentiation and maturation.

### 3.1. Pitx3 and Lmx1 a/b

LIM homeodomain transcription factors 1a and 1b (Lmx1a and Lmx1b) play key roles in the developing midbrain, where their expressions start on embryonic days 9.5 and 7.5, respectively [45,46]. Lmx1a is initially expressed in the ventral midbrain, and then gradually spreads to dorsal regions, whereas Lmx1b expression is constrained to several specific areas, including the ventral midbrain, floor plate, mid-hindbrain boundary, and roof plate [47]. The knockdown of Lmx1a expression leads to the loss of mDA neurons, whereas the activation of this gene promotes the generation of mDA neurons [48]. Additionally, DAT-Lmx1a/b^cKO^ adult mice, especially elderly mice, displayed motor, non-motor (olfaction), and cognitive impairments. Meanwhile, the model also showed lower striatal DA contents, altered synaptic structures at DA nerve terminals, and the progressive loss of mDA neurons and fibers [49].

Previous studies have reported that Lmx1a/b could directly regulate the expression of Nurr1 and Pitx3. Lmx1a binds directly to the murine Pitx3 promoter region in vivo, and retroviral Lmx1a expression increased the expression of Pitx3 in vitro. Moreover, Lmx1b is also bound to the promoters of Pitx3 and Nurr1, whereas in Lmx1b KO mice, the retained TH^+^ neurons all lack Pitx3 expression, indicating that Lmx1b is closely associated with Pitx3. Wnt1/β-catenin/LEF1 signaling plays an essential role in the development of the mouse’s ventral midbrain [50,51]. Lmx1a and Pitx3 are all potential direct targets of Wnt1/β-catenin/LEF1-mediated signaling [52], suggesting a cascade linking Wnt1/β-catenin/LEF1-Lmx1a-Pitx3 to mDA differentiation. Moreover, Lmx1b is also closely associated with Wnt1 signaling pathway, and the Wnt1-Lmx1b autoregulatory loop directly regulates Nurr1, En1, and Pitx3, all of which are crucial elements in mDA neuronal differentiation and survival [53,54]. Together, these observations suggest that Wnt1 signaling pathway is associated with Lmx1a/b and other important neural transcription factors (Nurr1, En1, and Pitx3) to participate in the early mDA developmental events.

Together, these observations suggest that Lmx1a/b and other important neural transcription factors (Nurr1, En1, and Pitx3) are associated with the Wnt1 signaling pathway to participate in the early mDA developmental events and Pitx3 could be directly regulated by Lmx1a/b.

### 3.2. Pitx3 and Nurr1

Orphan nuclear receptor Nurr1 (Nr4a2) is one of the main regulators for the development, survival, and functional maintenance of mDA neurons in vivo [10]. Nurr1 expression starts in the midbrain on embryonic day 10.5 and continues to be expressed in mDA neurons till adulthood [55]. Nurr1 is vital for establishing the dopaminergic phenotype in mDA neurons as it determines the expression of TH, the enzyme responsible for DA synthesis. In Nurr1-deficient mice, the expression of TH was completely absent during development, indicating that Nurr1 is essential for TH induction in mDA neurons [55,56]. Moreover, other important transcription factors (En1/2, FoxA2, Lmx1b, and Pitx3) were still retained in embryos; thus, no developmental abnormalities appeared in the mice. However, homozygous Nurr1 knock out mice died soon after birth, while heterozygous Nurr1^+/−^ mice lived longer, but their mDA neurons were more vulnerable to neurotoxic damage caused by MPTP injection or proteasome inhibition. Meanwhile, they also exhibited severe age-related mDA neurodegeneration [57]. According to these findings, Nurr1 has a minor role in the early development of mDA neurons, but it is essential for the late survival and differentiation of mDA neurons.

Recent research has demonstrated the importance of the Lmx1a-miR-204/211-Nurr1 axis in the differentiation of mDA neurons [58]. Additionally, mice that are overexpressed with mutant alpha-synuclein and simultaneously lacking one Nurr1 allele showed the enhanced age-dependent reduction of Nurr1 protein levels, a decreased number of mDA neurons, as well as increased neuroinflammation and alpha-synuclein aggregation [59]. The possible mechanism underlying these pathological phenotypes is that alpha-synuclein promoted the GSK3B-mediated phosphorylation (a component of the Wnt1/β-catenin pathway) and the subsequent proteasomal degradation of Nurr1 [60], suggesting the presence of a Wnt1/β-catenin-lmx1a-Nurr1 pathway in the differentiation of mDA neurons.

Nurr1 and Pitx3 are both expressed during mDA neuron differentiation. Nurr1 is involved in DA synthesis and regulation, while Pitx3 is specifically engaged in the terminal differentiation and maintenance of mDA neurons. Moreover, Nurr1 is widely distributed in the cerebral cortex, hippocampus, thalamus, amygdala, and midbrain [61]. By contrast, Pitx3 is postnatally expressed only by mDA neurons and is constrained in the SNc and VTA regions. Clinical studies showed that the expression levels of Nurr1 and Pitx3 were significantly decreased in PD patients [62]. In addition, the presence of Pitx3 and Nurr1 is essential for triggering TH expressions in mDA neurons. A growing body of evidence indicates that Pitx3 and Nurr1 are functionally interconnected and cooperate to promote the differentiation of mouse and human ES cells toward the mDA phenotype [63]. Recent studies have shown that Lentiviral expression vectors that combined harbor Nurr1 and Pitx3 at the neural precursor stage dramatically and synergistically induced the expression of the late marker (DAT), and Pitx3 serves as a crucial Nurr1 potentiator in defining the dopaminergic phenotype [64]. Pitx3 triggers the release of SMRT/HDAC (co-repressor)-mediated repression from Nurr1 to regulate the Nurr1 transcriptional complex [64]. Specifically, Pitx3 and Nurr1 co-occupy promoters of mDA-related genes, and Pitx3 causes HDAC-mediated repression in the Nurr1 transcriptional complex. The loss of Pitx3 expression in vivo increases the interaction of co-repressor SMRT with Nurr1, causing Nurr1-mediated transcriptions of Vmat2, D2R, and TH to be repressed in an HDAC-dependent manner [64]. Furthermore, Pitx3 and Nurr1 interact with the promoter of Vmat2, a gene essential for the metabolism of DA [10].

These findings imply that Pitx3 could enhance the function of Nurr1 and they cooperatively induce the late maturation of midbrain DA neurons.

### 3.3. Pitx3 and Engrailed-1/Engrailed-2

Engrailed-1/Engrailed-2 (En1/2), a member of a family of transcription factors containing a DNA-binding homeodomain, started to express on embryonic day 8 in a sheet of cells in the anterior neuroepithelium of midbrain–hindbrain boundary. Later on, the expression of En1/2 was restricted to the midbrain (SNc and VTA regions) and hindbrain regions in adults [65,66,67]. The progressive loss of striatal neurons and SNc mDA neurons with associated motor abnormalities has been identified in the En1/2-deficient models [68]. These data indicated that En1/2 plays an important role in the process of terminal differentiation and is required for the acquisition and survival of mature neuronal identity during late embryonic life in a dose-dependent manner [13,69]. En1/2 also interacts with Wnt1 to control the induction of the midbrain–hindbrain boundary (MHB), and knockdown Wnt1 downregulates En1/2 expression in the MHB region [70].

Pitx3 and En1 were reported to be involved in the initiation of the expression of aldehyde dehydrogenase 1 family member A1 (Aldh1a1), a gene that is an important marker in rostro-lateral mDA neurons [13,71]. A previous study indicated a strong association between the Pitx3 promoter and PD (*p* = 0.0001), as well as between EN1 and PD (*p* = 0.046), after genotyping nine single nucleotide polymorphisms in the entire genomic region of Pitx3 and EN1 from 365 PD patients and 418 controls [72]. Thus, En1 and Pitx3 polymorphisms may function as genetic risk factors for sporadic PD.

En1 knock out^−^ mice showed a significant loss of mDA neurons and striatal innervation defects, which were similar to those observed in Pitx3 knock out^−^ mice [71]. Additionally, on embryonic day 14.5, a small number of mDA neurons expressed Nurr1, but not Pitx3 and En1. These neurons are known to lose their rostral or caudal subset identity, suggesting the necessity of Pitx3 and En1 in the rostral/caudal identity of mDA neuronal populations [13]. Moreover, Pitx3 regulates En1 activity through multiple Pitx3-driven modulatory proteins, and En1 expression is increased in the absence of Pitx3 [71]. Meanwhile, En1 may be also involved in the regulation of Pitx3 expression, and the levels of Pitx3 were reduced in En1-deleted animals [13,71]. On the other hand, Pitx3 plays an essential role in the survival of rostro-lateral mDA neurons, but not caudo-lateral ones. Accordingly, in a Pitx3 knock out model, most of the rostro-lateral mDA neurons were lost, but the caudo-lateral ones commonly referred to as VTA mDA neurons were retained [40,71], whereas in En1 knock out embryos, not only rostral mDA neuronal programming is affected, but also caudal mDA phenotypes [71], suggesting that Pitx3 and En1 contribute to the modulation and specification of different subtypes of mDA neurons.

These findings illustrate that Pitx3 and En1 are both genetic risk factors for sporadic Parkinson’s disease. Moreover, there is a close relationship between the two, with En1 probably increasing Pitx3 expression, while Pitx3 might be suppressed by En1 expression.

### 3.4. Pitx3 and Foxa1/2

Foxa1/2, members of the winged-helix/forkhead transcription factors [73], are essential for the generation, specification, differentiation, and transcription of mDA neurons during embryonic development [74,75]. Foxa1 was identified in the floorplate, notochord, and endoderm on embryonic day 7.5, and upon Foxa1-deficiency, the mice suffered from disrupted glucose homeostasis and died soon after birth [76,77,78]. Foxa2 is expressed in the posterior epiblast from the early stage onwards, and it is then confined to anterior definitive endoderm (ADE) and axial mesoderm, which consists of the head process, prechordal plate, notochord, and node [76]. Foxa2 expression is necessary for proper mesodermal and endodermal development among mice, and it starts to express as early as embryonic day 6.5. The consecutive Foxa2 loss results in embryonic lethality [76,77,78]. Together, both Foxa1 and Foxa2 are expressed in nearly all mDA neurons in adult mouse brains and are required for normal feeding and gait behavior, as well as for maintaining the expressions of mDA-specific markers during late embryonic/early postnatal and adult stages [75,79].

Consistent with haploinsufficiency of En1/2 or adult deletion of Nurr1, the haploinsufficiency of *Foxa2* also results in the progressive loss of mDA neurons and decreased striatal DA, leading to the development of PD-related phenotypes in adult mice [16,80]. The co-expression and physical interaction of Foxa1/2 and Nurr1 have been identified in mDA neurons of adult mice, and the knockdown of Foxa1/2 could reduce the binding efficiency of Nurr1 to *TH* promoter regions [79,81]. Moreover, Foxa1/2 controls the expression of Lmx1a and Lmx1b (Lmx1a/b expression vanished in the Foxa1/2 knockout mutant) and is necessary for En1 and Nurr1 expression in developing mDA neurons [82]. It has been reported that Foxa1/2 could cooperate with Pitx3 and Nurr1 to enhance TH expression during late DA neuron development [64,82]. However, in Foxa1/2^cKO^ mice, neither Pitx3 nor Nurr1 expression were impacted in the SNc and VTA mDA neurons, suggesting that Foxa1/2 may enhance the function of Pitx3 rather than its expression, and other redundant pathways may contribute to the expression of Pitx3 and Nurr1 during neuronal development [75].

The above findings suggest that among the neural transcription factors mentioned in this paper, Foxa1/2 is located upstream of them and could regulate their expression, and in addition to Pitx3, Foxa1/2 may enhance its function more.

Taken together, we summarized the expression time and distribution regions of important transcription factors (Pitx3, Lmx1a/b, Nurr1, Engrailed 1/2, and Foxa1/2) for mDA neuron development in Table 1. Additionally, the involvement of nuclear transcription factors (Pitx3, Nurr1, Eng 1/2, and Foxa 1/2) in PD pathogenesis is shown in Figure 1, and the genetic regulation of transcription factors mentioned above in the development of the mDA neurons is detailed in Figure 2.

## 4. The Broad Role of Pitx3 in PD: From Model to Patients

A variety of rodent models of PD have been used to study the pathogenesis of the disease, and they can currently be divided into two basic categories: neurotoxin-induced and genetic models [84,85]. The former models are usually treated with paraquat, rotenone, 1-methyl-4-phenyl-1,2,3,6-tetrahydropyridine (MPTP), or 6-hydroxydopamine (6-OHDA) [86]; the latter models include those that overexpress α-synuclein or LRRK2 mutations, as well as DJ-1 knockout, PINK1 knockout, VMAT2 knockout, Nurr1 knockout, MitoPark knockout, Pitx3 knockout (aphakia) animal models, etc. [86,87]. These applied rodent models show highly suggestive human PD symptoms and are critical for understanding the motor/non-motor symptoms of PD [88].

### 4.1. Development of Pitx3-Deficient Mouse Models

Pitx3-deficiency was found in a natural mouse mutant (ophthalmic mice) that exhibited mDA neuron loss and motor abnormalities [89]. Unlike many genetic models, these mice selectively lost mDA neurons in the SNc, while mDA neurons in the ventral tegmental region and the olfactory bulb were largely retained, closely resembling the clinical aspects of PD [90], and the motor impairments of the mutants could be alleviated after levodopa injection [89]. Most intriguingly, these mice exhibited multiple non-motor deficits, such as an impaired spatial working memory, gastrointestinal dysfunction, and depressive behavior, which are analogous to the non-motor symptoms seen in PD patients. Meanwhile, inflammatory deficits were also identified in the Pitx3-deficient mice, including the morphologic alterations in astrocytes and microglia, and they potentially contributed to mDA neuron neurodegeneration as well [90]. Although the primary diagnostic criteria and most visible manifestations of PD are motor symptoms, non-motor symptoms are increasingly being acknowledged as an important component of PD dysfunctions and are frequently noticed before the onset of overt motor deficits [91]. A sleep disturbance, constipation, digestive dysfunction, and hyposmia have become important concerns for some PD patients [90]. Thus, Pitx3-deficient mice may provide some insight into the non-motor functional deficiencies of PD. Additionally, given that Pitx3 is an essential transcription factor for mDA development, Pitx3-deficient mice represent a useful tool that permits the study of early events in mDA formation. Moreover, studies have reported that Pitx3^−/−^ mice produce decreased DA signaling rates and induce motor deficits [92] and can be also used to study the morphological and physiological modulation of striatal neurons [93] even after the administration of L-DOPA or other PD-related drugs [94].

Our laboratory recently generated a line of Pitx3^fl/fl^/DAT^CreERT2^-conditional knockout mouse models that postnatally knocks out the Pitx3 gene in the cells expressing the dopamine transporter (DAT) protein after a tamoxifen (TAM) treatment. The mouse model showed canonical neuropathological features of PD, including age-dependent progressive motor deficits, a sharp decline in striatal DA content, and a significant loss of mDA neurons in the SNc, but not in the VTA [42], indicating the importance of the Pitx3 gene in adult neuronal survival. Moreover, Pitx3 deficiency also contributes to the pathology of the striatum. Our studies have shown that the contents of multiple neurotransmitters decreased in the Pitx3-deficient mice at early stages, including DA, GABA, and glutamate. Meanwhile, the morphologies of medium spiny neurons (MSNs) were altered due to Pitx3 deficiency, including nuclear, soma, and dendritic atrophy, as well as an increased number of nuclear invaginations. Additionally, more nuclear DNA damages were observed in MSNs during aging, while Pitx3-deficiency aggravated this phenomenon, together with alterations of the DNA methylation profiling associated with lipoprotein and nucleus pathways at late stages [42,93], suggesting that early perturbations of neurotransmitters within MSNs may potentially contribute to the alterations of metabolism, morphology, and epigenetics within the striatum at late stages.

### 4.2. The Role of Pitx3 in Cell Therapy

Cell transplantation, a potential strategy used to replace impaired mDA neurons, has gained particular interest since the 21st century. Stem cells have unique qualities, with the capacity to self-renew and differentiate into various cell types [95]. An increasing body of evidence supports that a series of transcription factors essential for mDA neuron development, differentiation, and specification also contribute to cell reprogramming: the generation of functional mDA neurons from stem cells or other non-neuronal cells [96].

Human teratocarcinoma cell line Ntera2 (NT2), which is derived from a human testicular cancer, is a potential source of cell therapy for neurological diseases [97]. NT2-derived neuron-like cells can give rise to the growth of axons and dendrites [98]. Thus, NT2 cells have long been thought of as a replacement source of mDA neurons for PD cell treatment [99]. Recent studies showed that after overexpressing Pitx3 and being exposed to GDNF, NT2 cells could release DA, indicating that Pitx3-GDNF interactions in DA signaling may promote the dopaminergic neuronal properties of NT2 cells, making it clinically applicable for cell replacement therapy in PD [99].

Mesenchymal stem cells (MSCs) derived from adult human bone marrow have varying capacities for brain development depending on the tissue of origin [100] and have shown lots of potential for immediate clinical applications due to their low immunogenicity [101]. However, a limited degree of mDA differentiation was seen in MSCs with a poor graft survival after transplantation [102]. Especially, adult bone marrow MSCs obtained from an older donor exhibited a significantly reduced neuroectodermal differentiation capability due to aging [103]. Pitx3 has been known to be a selective marker for transplantable embryonic stem cells [104,105]. The overexpression of Pitx3 in MSCs promotes the induction of mDA phenotype [25], suggesting that the administration of Pitx3 in MSCs during transplantation may be an effective cell therapy in PD. However, more research may be required to find the most suitable source of MSCs and to standardize the production process for ensuring the uniform neural differentiation of MSCs.

Research has shown that only 5–10% of neural stem cells survive after transplantation due to the toxic effects of the inflammatory state. Meanwhile, the majority of neural stem cells transplanted in vivo differentiate into glial cells rather than neurons [106,107]. Thus, it is crucial to find a way to prevent neuroinflammation in mDA neurons and promote their survival and development. Applying developmental transcription factors to therapeutics has become one of the most popular and appealing strategies [16]. Exogenous Pitx3 in ESCs-derived progenitor cells promotes the development of DA neurons in vitro by controlling the expression of genes for TH, Ngn2, and tubulin III [108], and an increased Pitx3 expression level was substantially linked with high TH expression [109] and benefits the formation of functional mDA neurons.

### 4.3. Pitx3 and PD

Polymorphisms of the Pitx3 gene are associated with the sporadic and early onset of PD [25,26,29]. One genetic association study from a screening sample of 340 PD patients and 680 controls and a large replication sample of 669 PD patients and 669 controls found the C allele of the Pitx3 promoter SNP rs3758549: C>T (*p* = 0.004) in PD patients appeared to be recessive with an estimated population frequency of 83%, suggesting an allele-dependent dysregulation of Pitx3 expression might contribute to the susceptibility to PD [110]. In contrast to abovementioned study, another study indicated that based on their clinical results, SNP is not confirmed to be associated with PD [111]. Additionally, an allele of rs4919621 SNP (chr10:103988621) in the intron of the Pitx3 gene and the C allele of rs2281983 SNP appeared significantly more often in PD patients with an early age of onset than they did in PD patients who experienced late onset (*p* = 0.001) and the controls (*p* = 0.002) [111]. Moreover, a significantly lower expression level of Pitx3 was recognized in the peripheral blood lymphocytes of PD patients compared to that of the controls. Importantly, this reduction was remarkably associated with an increased risk of developing PD in male and elderly subjects [28]. The data indicate that variations in transcription factors, which are crucial for the growth and maintenance of mDA neurons, merits our attention as they may be directly associated with PD vulnerability. To increase the precision and reproducibility of experimental outcomes, the onset of PD (early or late onset) and whether there is a PD family history in PD patients should also be taken into account.

At present, the primary therapeutic strategy used to alleviate or control motor symptoms in PD patients is to generally increase the bioavailability of DA by supplementing DA precursors or inhibiting DA breakdown [112]. However, as the disease progresses and the continuous loss of SNc mDA neurons occurs, the treatment efficiency decreases, and some of the common side effects of a long-term levodopa treatment occur (motor fluctuations, “on/off” phenomena, and dyskinesias). These levodopa-related complications and disabilities have become a treatment challenge for patients with advanced PD [14]. It has been reported that the delivery of pro-survival/anti-apoptotic factors to PD patients either locally (e.g., into the brain/VM or directly into mDA neurons) or systemically (e.g., into the bloodstream) is not sufficient to create efficient PD prevention and neuroprotective therapies. The possible reasons may be the timing and manner of administration, as well as potential concentration–dependent interactions, i.e., side effects in the body or brain, must be taken into account [113]. In our Pitx3^cKO^ mouse model, a profound loss of mDA neurons in SNc was identified; however, the proximal neurons within VTA are less vulnerable to degeneration and are largely spared [42]. This means that different mDA neuron subtypes own their distinct vulnerability to the PD pathology. Therefore, our model provides a useful tool for elucidating how neuronal specification is maintained in subtypes of adult mDA neurons, which contributes to the development of timing and manner of drug administration in PD.

Furthermore, although Pitx3 is not expressed in astrocytes, the transfection of Pitx3 in astrocytes increased the expression levels of BDNF and GDNF, resulting in a protective effect on mDA neurons. Specifically, Pitx3-transfected astrocytes greatly mitigate rotenone-induced damage to mDA neurons, whereas this protection can be significantly reversed by preincubation with antibodies against BDNF or GDNF [114]. These results suggested that Pitx3 may protect mDA neurons by being overexpressed in non-mDA neurons. Thus, the co-transplantation of non-neuronal cells overexpressing Pitx3 with MSCs may increase graft survival and integration, but this still requires further study in the future. Studies have shown that genetic factors affect patients more during early-onset PD than they do during late-onset PD. Likely, Pitx3 is also critical for early-onset PD [115]. Therefore, the stage of PD onset is a key factor that needs to be considered when performing PD treatments associated with Pitx3. Moreover, further unraveling the complexity of PD pathogenesis and expanding the knowledge of the synergistic effects between transcription factors in mDA neuron development will contribute to promoting the development of alternative therapeutic strategies for PD.

## 5. Conclusions and Perspective

A better understanding of neural networks of mDA-related transcription factors could help to enhance the prevention or improve the treatments of PD. Pitx3 is a crucial transcription factor for the growth of mDA neurons and promotes the differentiation and survival of SNc mDA neurons. It has been reported that Pitx3 associates with multiple transcription factors (Lmx1 a/b, Nurr1, En1/2, and Foxa 1/2) and cooperate with them to participate in the differentiation and expansion of mDA neurons. Therefore, Pitx3 may be an entry point to regulate pathways essential for the prevention and treatment of PD. We believe that increasing the knowledge of Pitx3 will enhance the mechanistic understating, drug development, and clinical evaluation of PD.

## Figures and Tables

**Figure 1 ijms-24-08614-f001:**
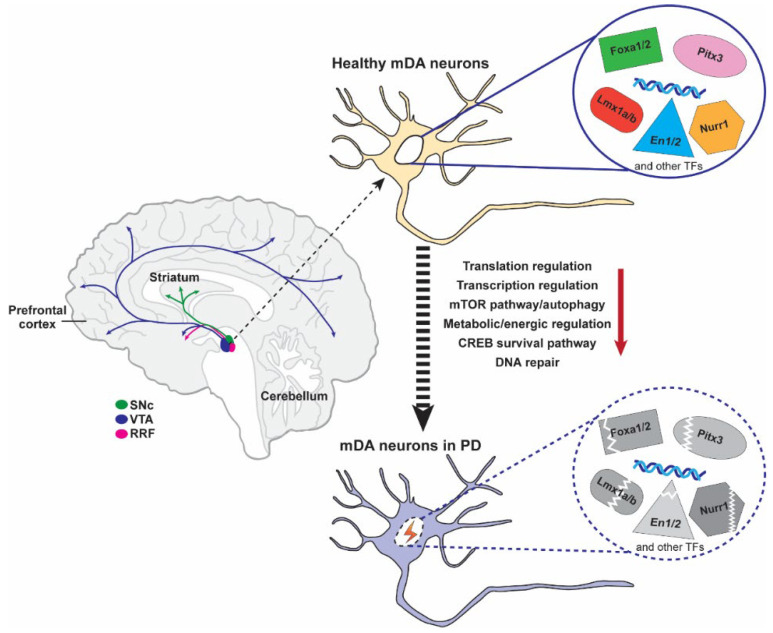
Projections of mDA neurons and involvement of nuclear transcription factors (Pitx3, Nurr1, Eng 1/2, and Foxa 1/2) in PD pathogenesis. A sagittal section of the adult brain shows the location of the cell bodies (ovals) of the RRF (pink), SNc (green), and VTA (blue) mDA clusters in the brain stem and their projection (indicated by arrows) into the limbic region (midbrain pathway), striatum (substantia nigra pathway), and PFC (midcortical pathway). Under physiological conditions, these neurotranscription factors (Pitx3, Nurr1, Eng 1/2, and Foxa 1/2) function normally. However, the dysfunction of these neurotranscription factors is accompanied by a series of abnormal indicators in the brain of PD patients, including abnormal transcription and regulation, an impaired mTOR pathway/autophagic pathway, the CREB survival pathway, and DNA repair under various stimuli (neurotoxin, aging, oxidative stress, etc., as indicated by the red lightning bolt icon), which ultimately leads to the death of mDA neurons.

**Figure 2 ijms-24-08614-f002:**
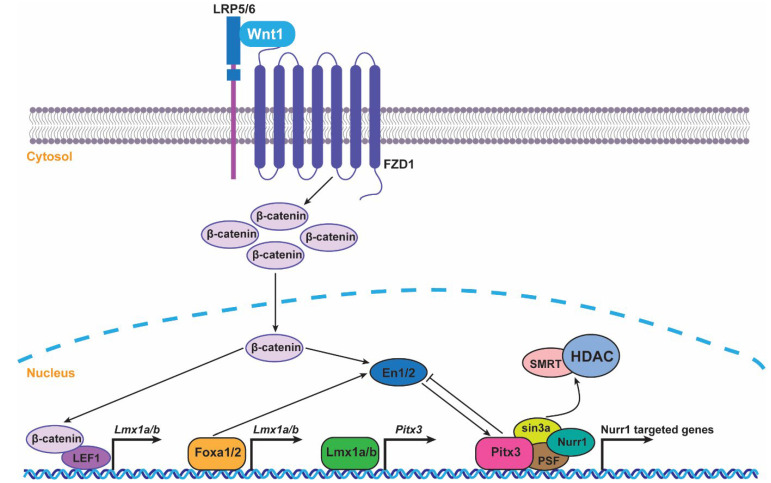
Genetic regulation of transcription factors in the development of the mDA neurons. Wnt1 interacts with Lmx1a/b, forming a Wnt-Lmx1a/b loop. The specific signaling pathways include WNT/β-catenin/LEF1-Lmx1a/b-Pitx3. Lmx1a/b is also regulated by Foxa1/2, which could control the expression of En1/2. En1/2 regulates Pitx3 expression and Pitx3 suppress En1/2. Pitx3 plays a critical role in SMRT-Sin3-HDAC-mediated repression of mDA-related genes by recruitment of the Nurr1-PSF transcriptional complex and releasing SMRT-HDAC complexes to activate Nurr1 transcription.

**Table 1 ijms-24-08614-t001:** The expression time and distribution region of important transcription factors for mDA neuron development.

Transcription Factor	The Expression Time Embryonic Day (E)	The Cells It Acts on	The Distribution Region	Reference
Pitx3	E11.5	Mature mDA neurons	Substantia nigra and ventral tegmental region	[23,61]
Lmx1a	E9	Mitotic ventral mesencephalic precursor cells	Ventral midbrain and dorsally	[46,47]
Lmx1b	E7.5	Neural progenitor cells, Mitotic ventral mesencephalic precursor cells	Derive from the midbrain, then restricted to the ventral midbrain, the floor plate, mid-hindbrain boundary, and roof plate	[46,47]
Nurr1	E10.5	Immature/ mature mDA neurons	Cerebral cortex, hippocampus, thalamus, amygdala, and midbrain	[61,83]
En1/2	E8	Ventral mDA precursor	Midbrain and hindbrain	[65,66,67]
Foxa 1	E7.5	mDA progenitors, immature/ mature neurons	Floorplate, notochord, and endoderm	[76,77,79]
Foxa 2	E6.5	mDA progenitors, immature/ mature neurons	Posterior epiblast, anterior definitive endoderm, axial mesoderm	[76,77,79]

## Data Availability

Not applicable.

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
