# Peer review of "The Crucial Roles of Pitx3 in Midbrain Dopaminergic Neuron Development and Parkinson’s Disease-Associated Neurodegeneration"

_ijms, 2023, doi:10.3390/ijms24108614_

Round 1

Reviewer 1 Report

This manuscript describes the role of Pitx3 in the development of Parkinson's Disease (PD). I think the manuscript is well written. Below are my comments for this review article.

1) Section 2 seems completely out of place to me. While the authors have tried to elaborate the interactions between Pitx3 and other transcriptional networks, most of the sections end up describing only the transcription factors and do not elaborate much on Pitx3.

2) I would like to see a more detailed background on Pitx3 and its role in PD. While these points are highlighted, I believe for a paper that is based on Pitx3, I think its role and functions are not emphasized enough.

Author Response

Reviewer #1: This manuscript describes the role of Pitx3 in the development of Parkinson's Disease (PD). I think the manuscript is well written. Below are my comments for this review article.

Response: We are very grateful for the positive feedbacks from the reviewer, The parts that need to be added and modified have been highlighted.

  1. Section 2 seems completely out of place to me. While the authors have tried to elaborate the interactions between Pitx3 and other transcriptional networks, most of the sections end up describing only the transcription factors and do not elaborate much on Pitx3.

Response: Appreciate the reviewer’s comments. We have made extensive revisions to further elucidate the linkage between Pitx3 and other transcription factors in section 2. The revised parts from lines 188-190, lines 201-204, lines 240-242, lines 239-241, lines 251-252, lines 287-289, lines 314-319 have been highlighted.

  1. I would like to see a more detailed background on Pitx3 and its role in PD. While these points are highlighted, I believe for a paper that is based on Pitx3, I think its role and functions are not emphasized enough.

Response: Appreciate the reviewer’s comments. We have added a new paragraph in the Background section (lines 106-119) and a new section of Introduction of Pitx3 (lines 128-166) as follows to further describe the background of Pitx3 and its role in PD:

“Among them, Pitx3 is mainly expressed in mDA neurons and plays an essential role in mDA neuronal development [23] [24]. The expression of Pitx3 is known to initiate as early as embryonic day 11.5, and is distributed in the lens, skeletal muscles, and mDA neurons during early development [23, 25, 26]. However, postnatally its expression is preserved only in postmitotic mDA precursors and mature mDA neurons particularly in SN, suggesting a key role for Pitx3 in the maintenance of mDA neuronal function [23, 24, 27]. Pitx3-deficient Aphakia (ak) mice exhibit preferential loss of SNc mdDA neurons during embryonic and postnatal development, reiterating that Pitx3 plays a critical role in early mdDA neuronal events. Additionally, the expression of Pitx3 was reported to be significantly downregulated in both PD patients' peripheral blood lymphocytes and brain tissues [25, 28]. Also, several Pitx3 gene variants have been identified to be linked to sporadic PD [29, 30]. Thus, these data altogether establish the key role of Pitx3 not only in early mDA development but also in retaining the normal physiological functions in adult neurons.”

2. Introduction of Pitx3

Pitx3, a member of the pituitary homeobox subfamily, is a transcription factor containing homeodomains located on chromosome 10q24 [33]. The first member of the pituitary homeobox family to be cloned was Pitxl (previously known as Ptx1). It was found to support pituitary development and function [34]. Shortly after a second family member, Pitx2 (called Ptx2 or Rieg1), was discovered to be involved with Rieger's syndrome, a deformity of the craniofacial characteristics [35] Then Pitx3 (Plx3) was identified during embryogenesis. It was momentarily expressed in skeletal muscle and the eye lens [23], but after birth, Pitx3 has highly restricted and constitutive expression in the SNc and VTA of the midbrain [23].

2.1.Pitx3 structure

In humans, Pitx3 has four exons and is found on chromosome 10. The protein is 302 amino acids long and the cDNA is 1407 bp in length [26]. The mouse Pitx3 gene also has four exons and is located on chromosome 19. The cDNA has a length of 1379 bp and the protein is also 302 amino acids long [25, 36]. The pitx3 genes of humans and mice are highly homologous and nearly 90% identical at the nucleotide level throughout the coding region [36].

2.2. Pitx3 Protein

Pitx3 coordinates gene activity that determines cell destiny during development in a variety of species, including yeasts and mammals [25]. The protein has an OAR domain at its C-terminus, which is present in several types of paired-like homeodomains [31]. Although the function of OAR in the Pitx3 protein is uncertain, OAR's impact on DNA binding appears to limit the transcriptional activity of the Cartl protein. The OAR domain and N-terminus of the Pitx and Prx OAR-related proteins are thought to interact intramolecularly, resulting in a protein conformation that is connected to a generally inactive state of these transcription factors [37].

2.3. Pitx3 in the main subgroups of mDA neurons

Pitx3 is a transcription factor required for the development and survival of mdDA neurons [38]. In the midbrain, DA neurons have two primary subgroups: the SNc and the VTA [7]. Although most mDA neurons express Pitx3, the Pitx3 deficiency in mice only impacts a portion of the mDA neurons. These findings show that Pitx3 may play distinct roles in these two midbrain subregions [25]. Pitx3 and TH are both expressed in the VTA at approximately the same time, and TH is still expressed in VTA even in the absence of Pitx3. However, in SNc, Pitx3 is expressed before TH, and the SNc precursors are unable to generate TH without Pitx3 [39]. Thus, the VTA DA neurons in Pitx3-deficient newborn mice remain unaffected, whereas, SNc neurons have been significantly lost potentially due to the decreased TH levels [40, 41]. These findings imply that Pitx3 is crucial for maintaining normal function and preserving the specific molecular identities of postnatal mDA neurons [42].”

Reviewer 2 Report

Based on the information provided in the manuscript, Pitx3 is a transcription factor that plays a crucial role in the development and survival of midbrain dopaminergic (mDA) neurons, which are the cells that degenerate in Parkinson's disease (PD). The transcription factor promotes the differentiation and survival of SNc mDA neurons and is also associated with multiple other transcription factors, including Lmx1 a/b, Nurr1, En1/2, and Foxa 1/2. Additionally, Pitx3 has been shown to have a neuroprotective effect on mDA neurons when overexpressed in non-mDA neurons such as astrocytes. Pitx3 is a promising target for PD treatment. Analyzing in detail the manuscript, there are a number of improvement suggestions I can have:

1. The introduction could benefit from more context about Parkinson's disease and its current treatments, as well as a brief overview of the role of transcription factors in the development and function of neurons.

2. The physiology of mDA neurons could be detailed to improve the introduction, as these cells represent one of the article's main subjects.

3. Describing the role of Pitx3 in the development and maintenance of mDA neurons will help the reader to enter into the subject. Not all readers have the professional background to follow a highly scientific-technical paper. Sometimes the information needs to be translated into other scientific fields, so simplifying the language and providing more examples or diagrams could help readers unfamiliar with the topic.

4. Adding more information about the genetics and molecular mechanisms of Pitx3 could make the section more informative and interesting. It could be explained, more precisely, the expression time, embryonic day 11,5. and which are the early events in mDA formation (line 271).

5. More detail could be provided about the specific genetic variants of Pitx3 that are associated with PD, as well as the strengths and limitations of the evidence for these associations.

6. The limitations and challenges of studying Pitx3 and PD, such as the complexity of the neural networks involved and the variability of patients and disease course, could be acknowledged and discussed in more detail.

7.  The conclusion could be more concise and specific in summarizing the main points of the article and highlighting the implications for future research and clinical applications.

Author Response

Reviewer #2: Based on the information provided in the manuscript, Pitx3 is a transcription factor that plays a crucial role in the development and survival of midbrain dopaminergic (mDA) neurons, which are the cells that degenerate in Parkinson's disease (PD). The transcription factor promotes the differentiation and survival of SNc mDA neurons and is also associated with multiple other transcription factors, including Lmx1 a/b, Nurr1, En1/2, and Foxa 1/2. Additionally, Pitx3 has been shown to have a neuroprotective effect on mDA neurons when overexpressed in non-mDA neurons such as astrocytes. Pitx3 is a promising target for PD treatment. Analyzing in detail the manuscript, there are a number of improvement suggestions I can have:

  1. The introduction could benefit from more context about Parkinson's disease and its current treatments, as well as a brief overview of the role of transcription factors in the development and function of neurons.

Response: As requested by the reviewer, we added a new paragraph about Parkinson's disease and its current treatments, as well as a brief overview of the role of transcription factors in the development and function of neurons in the Background section (lines 86-105) as follows:

“Currently, the main PD treatments still focus on increasing DA level or regulating DA transmission by pharmacotherapy [14]. Moreover, patients who have noticeable tremors or uncontrollable motor fluctuations may also benefit from deep brain stimulation operation [15]. However, instead of preventing or delaying the progression of the disease, these treatments can only partially mitigate the physical symptoms [16]. Meanwhile, substantial side effects, like dyskinesia and impulse control difficulties, may appear after long-term pharmacological treatments [17] Cell transplantation, a potential strategy to replace impaired mDA neurons, has gained particular interest. It has been known that some transcription factors are crucial for the growth, differentiation, specification, and transmitter synthesis of mDA neurons, including forkhead box A2 (Foxa2), LIM homeobox transcription factor1 A (Lmx1a), nuclear receptor-related factor1 (Nurr1), paired-like homeodomain3 (Pitx3), aldehyde Dehydrogenase 1 family member A1 (Ascl1), neurogenin2 (Ngn2), and neuronal differentiation 1 (NeuroD1) [16]. Pitx3 and Nurr1 are mdDA neuron-specific markers, whereas Ascl1, Ngn2, and NeuroD1 are markers for neural progenitor cells. Additionally, Neurotrophic factors including glial-cell-derived neurotrophic factor (GDNF) and brain-derived neurotrophic factor (BDNF) also play a crucial part in the differentiation, development, and maintenance of mDA [16, 18-22]. Thus, expanding the knowledge of these transcription factors and neurotrophic factors in neuronal formation will greatly benefit the generation of functional mDA neurons that are applied for cell replacement therapy in PD patients.”

  1. The physiology of mDA neurons could be detailed to improve the introduction, as these cells represent one of the article's main subjects.

Response: Appreciate the reviewer’s comments. We have modified the paragraph to elucidate the physiology of mDA neurons in the Background section (lines 61-73) as follows:

“The mDA neurons are generally divided into three main clusters, located in substantia nigra pars compacta (SNc), ventral tegmental area (VTA), and the retrorubral field (RRF). SNc mDA neurons primarily project their axons to dorsolateral striatum, a component of the basal ganglia circuitry, and DA released from SNc mDA neurons mediates the activity of targeted striatal cells to regulate voluntary movements and motor learning [11]. In contrast, VTA mDA neurons mainly innervate ventromedial striatum (nucleus accumbens, NAc), prefrontal cortex (PFC; mesocortical pathway), and limbic regions (mesolimbic network). DA release from VTA and RRF mDA neurons are involved in the modulation of cognitive, emotive/affective, motivational/salient and rewarding behaviors [11]. All three types of mDA neurons undergo multiple developmental stages, including patterning, induction, specification of mDA progenitors, and mDA neuron maturation, which are controlled by numerous genetic networks and cellular signaling pathways [12] [13].”

  1. Describing the role of Pitx3 in the development and maintenance of mDA neurons will help the reader to enter into the subject. Not all readers have the professional background to follow a highly scientific-technical paper. Sometimes the information needs to be translated into other scientific fields, so simplifying the language and providing more examples or diagrams could help readers unfamiliar with the topic.

Response: Appreciate the reviewer’s comments. We have added a new paragraph in the Background section (lines 106-119) and a new section of Introduction of Pitx3 (lines 128-165) as follows to describe the the role of Pitx3 in the development and maintenance of mDA neurons:

“Among them, Pitx3 is mainly expressed in mDA neurons and plays an essential role in mDA neuronal development [24] [25]. The expression of Pitx3 is known to initiate as early as embryonic day 11.5 and is distributed in the lens, skeletal muscles, and mDA neurons during early development [24, 26, 27]. However, postnatally its expression is preserved only in postmitotic mDA precursors and mature mDA neurons particularly in SN, suggesting a key role for Pitx3 in the maintenance of mDA neuronal function [24, 25, 28]. Pitx3-deficient Aphakia (ak) mice exhibit preferential loss of SNc mdDA neurons during embryonic and postnatal development, reiterating that Pitx3 plays a critical role in early mdDA neuronal events. Additionally, the expression of Pitx3 was reported to be significantly downregulated in both PD patients' peripheral blood lymphocytes and brain tissues [26, 29]. Also, several Pitx3 gene variants have been identified to be linked to sporadic PD [30, 31]. Thus, these data altogether establish the key role of Pitx3 not only in early mDA development but also in retaining the normal physiological functions in adult neurons.”

2. Introduction of Pitx3

Pitx3, a member of the pituitary homeobox subfamily, is a transcription factor containing homeodomains located on chromosome 10q24 [33]. The first member of the pituitary homeobox family to be cloned was Pitxl (previously known as Ptx1). It was found to support pituitary development and function [34]. Shortly after a second family member, Pitx2 (called Ptx2 or Rieg1), was discovered to be involved with Rieger's syndrome, a deformity of the craniofacial characteristics [35] Then Pitx3 (Plx3) was identified during embryogenesis. It was momentarily expressed in skeletal muscle and the eye lens [23], but after birth, Pitx3 has highly restricted and constitutive expression in the SNc and VTA of the midbrain [23].

2.1.Pitx3 structure

In humans, Pitx3 has four exons and is found on chromosome 10. The protein is 302 amino acids long and the cDNA is 1407 bp in length [26]. The mouse Pitx3 gene also has four exons and is located on chromosome 19. The cDNA has a length of 1379 bp and the protein is also 302 amino acids long [25, 36]. The pitx3 genes of humans and mice are highly homologous and nearly 90% identical at the nucleotide level throughout the coding region [36].

2.2. Pitx3 Protein

Pitx3 coordinates gene activity that determines cell destiny during development in a variety of species, including yeasts and mammals [25]. The protein has an OAR domain at its C-terminus, which is present in several types of paired-like homeodomains [31]. Although the function of OAR in the Pitx3 protein is uncertain, OAR's impact on DNA binding appears to limit the transcriptional activity of the Cartl protein. The OAR domain and N-terminus of the Pitx and Prx OAR-related proteins are thought to interact intramolecularly, resulting in a protein conformation that is connected to a generally inactive state of these transcription factors [37].

2.3. Pitx3 in the main subgroups of mDA neurons

Pitx3 is a transcription factor required for the development and survival of mdDA neurons [38]. In the midbrain, DA neurons have two primary subgroups: the SNc and the VTA [7]. Although most mDA neurons express Pitx3, the Pitx3 deficiency in mice only impacts a portion of the mDA neurons. These findings show that Pitx3 may play distinct roles in these two midbrain subregions [25]. Pitx3 and TH are both expressed in the VTA at approximately the same time, and TH is still expressed in VTA even in the absence of Pitx3. However, in SNc, Pitx3 is expressed before TH, and the SNc precursors are unable to generate TH without Pitx3 [39]. Thus, the VTA DA neurons in Pitx3-deficient newborn mice remain unaffected, whereas, SNc neurons have been significantly lost potentially due to the decreased TH levels [40, 41]. These findings imply that Pitx3 is crucial for maintaining normal function and preserving the specific molecular identities of postnatal mDA neurons [42].”

  1. Adding more information about the genetics and molecular mechanisms of Pitx3 could make the section more informative and interesting. It could be explained, more precisely, the expression time, embryonic day 11,5. and which are the early events in mDA formation (line 271).

Reply: Appreciate the reviewer’s comments. More information about the genetics and molecular mechanisms of Pitx3 could be found in the section 2 “Introduction of Pitx3” (just above, in the question #3) we have newly added. Additionally, we also added a new paragraph in the Background section (lines 46-60) as follows to describe the early events in mDA formation.

“mDA neurons in the SNc and VTA of the midbrain are the primary DA source in the mammalian central nervous system [7]. During development, the precursors of DA neurons migrate from the neuroepithelium to the ventral midbrain. Sonic hedgehog (Shh) and fibroblast growth factor (Fgf8), two diffusible factors, collaborate to promote neuronal development. Lmx1a and Msx1 appear early while the precursor cells are proliferating, and these are essential for the passage of the proliferating precursor cells into the first stages of differentiation into mDA neurons [9]. Lmx1b, the engrailed factors En1 and En2, Foxal, and other factors are also involved in the early stage of mDA-specific differentiation (at E9-E10 in mice), when the first gene for DA synthesis, amino acid decarboxylase (AADC) is induced. Between E9.5 and E13.5 in mice, midbrain DA progenitors exit the cell cycle and enter post-mitosis [10]. Then, at E10.5 and E11.5, the transcription factors for terminal differentiation are activated by Nurr1 and Pitx3, respectively. Nurr1 is required for the induction of tyrosine hydroxylase (TH) at E11.5, vesicular monoamine transporter 2 (Vmat2) at E12.5, and dopamine transporter (DAT) at E14 [11], and Pitx3 is necessary for some terminally differentiated SNc DA neurons to survive [7].”

  1. More detail could be provided about the specific genetic variants of Pitx3 that are associated with PD, as well as the strengths and limitations of the evidence for these associations.

Reply: As requested by the reviewer, we revised the section 4.3. Pitx3 and PD (lines 422-440) as follows:

“Polymorphisms of the Pitx3 gene are associated with sporadic and early onset of PD [25, 26, 29]. One genetic association study from a screening sample of 340 PD patients and 680 controls, and a large replication sample of 669 PD patients and 669 controls found the C allele of the Pitx3 promoter SNP rs3758549: C>T (p=0.004) in PD patients appeared to be recessive with an estimated population frequency of 83%, suggesting an allele-dependent dysregulation of Pitx3 expression might contribute to the susceptibility to PD [110]. In contrast to above mentioned, another study indicated that based on their clinical results, such SNP is not confirmed to be associated with PD [111]. Additionally, an allele of the rs4919621 SNP (chr10:103988621) in the intron of the Pitx3 gene and the C allele of the rs2281983 SNP appeared significantly more often in PD patients with an early age of onset than PD patients with late-onset (p=0.001) and controls (p=0.002) [111]. Besides, a significantly lower expression of Pitx3 was recognized in the peripheral blood lymphocytes of PD patients compared to controls. Importantly such reduction was remarkably associated with an increased risk of developing PD in male and elderly subjects [28]. The data indicate that variation in transcription factors, which are crucial for the growth and maintenance of mDA neurons, merits our attention as it may be directly associated with PD vulnerability. To increase the precision and reproducibility of experimental outcomes, the onset of PD (early or late onset) and whether there is a PD family history in PD patients should also take into account.”

  1. The limitations and challenges of studying Pitx3 and PD, such as the complexity of the neural networks involved and the variability of patients and disease course, could be acknowledged and discussed in more detail.

Response: Appreciate the reviewer’s comments. This question is related to the question #5. Thus, the reviewer could check the answers above. Additionally, we also added new paragraphs in the section 4.2. The role of Pitx3 in cell therapy (lines 411-420) and 4.3. Pitx3 and PD (lines 460-474) as follows for further describing the limitations and challenges of studying Pitx3 and PD :

“Research has shown that only 5–10% of neural stem cells survive after transplantation due to the toxic effects of the inflammatory state. Meanwhile, the majority of neural stem cells transplanted in vivo differentiate into glial cells rather than neurons [106, 107]. Thus, it is crucial to find a way to prevent neuroinflammation in mDA neurons and promote their survival and development. Applying developmental transcription factors in therapeutics has become one of the most popular and appealing strategies [16]. Exogenous Pitx3 in ESCs-derived progenitor cells promotes the development of DA neurons in vitro by controlling the expression of the genes for TH, Ngn2, and -tubulin III [108], and increased Pitx3 expression was substantially linked with high TH expression [109] and benefits the formation of functional mDA neurons.”

“Furthermore, although Pitx3 is not expressed in astrocytes, transfection of Pitx3 in astrocytes increased the expressions of BDNF and GDNF, resulting in a protective effect on mDA neurons. Specifically, Pitx3-transfected astrocytes greatly mitigate rotenone-induced damage to the mDA neurons. Whereas, this protection can be significantly reversed by preincubation with antibodies against BDNF or GDNF [114]. These results suggested that Pitx3 may protect mDA neurons by being overexpressed in non-mDA neurons. Thus, co-transplantation of non-neuronal cells overexpressing Pitx3 with MSCs may increase graft survival and integration, but that still requires further study in the future. Studies have shown that genetic factors affect more during early-onset PD than late-onset PD. Likely, Pitx3 is also critical for early-onset PD [115]. Therefore, the stage of PD onset is a key factor that needs to be considered when performing the PD treatments associated with Pitx3. Moreover, further unraveling the complexity of PD pathogenesis and expanding the knowledge of the synergistic effects between transcription factors in mDA neuron development will contribute to promoting the development of alternative therapeutic strategies for PD.”

  1. The conclusion could be more concise and specific in summarizing the main points of the article and highlighting the implications for future research and clinical applications.

Reply: Appreciate the reviewer’s comments. We have revised the conclusion part (lines 475-483) as follows:

“A better understanding of neural networks of mDA-related transcription factors could help to enhance the prevention or improve the treatments of PD. Pitx3 is a crucial transcription factor for the growth of mDA neurons and promotes the differentiation and survival of SNc mDA neurons. It has been reported that Pitx3 associates with multiple transcription factors (Lmx1 a/b, Nurr1, En1/2, and Foxa 1/2) and cooperate with them to participate in the differentiation and expansion of mDA neurons. Therefore, Pitx3 may be an entry point to regulate pathways essential for the prevention and treatment of PD. We believe that increasing the knowledge of Pitx3 will enhance the mechanistic understating, drug development, and clinical evaluation of PD.”

Round 2

Reviewer 2 Report

The authors have addressed the comments and suggestions in their responses to the previous evaluation report. I conclude for Accept in present form